# A vitamin-B2-sensing mechanism that regulates gut protease activity to impact animal's food behavior and growth

Bin Qi, Marina Kniazeva, Min Han*

Department of Molecular, Cellular and Developmental Biology, Howard Hughes Medical Institute, University of Colorado Boulder, Boulder, United States

**Abstract** To survive challenging environments, animals acquired the ability to evaluate food quality in the intestine and respond to nutrient deficiencies with changes in food-response behavior, metabolism and development. However, the regulatory mechanisms underlying intestinal sensing of specific nutrients, especially micronutrients such as vitamins, and the connections to downstream physiological responses in animals remain underexplored. We have established a system to analyze the intestinal response to vitamin $B_2$ (VB2) deficiency in *Caenorhabditis elegans*, and demonstrated that VB2 level critically impacts food uptake and foraging behavior by regulating specific protease gene expression and intestinal protease activity. We show that this impact is mediated by TORC1 signaling through reading the FAD-dependent ATP level. Thus, our study in live animals uncovers a VB2-sensing/response pathway that regulates food-uptake, a mechanism by which a common signaling pathway translates a specific nutrient signal into physiological activities, and the importance of gut microbiota in supplying micronutrients to animals.

*For correspondence: mhan@colorado.edu

**Competing interests:** The authors declare that no competing interests exist.

## Introduction

To survive challenging environments with fluctuating nutrient resources, animals have acquired the ability to evaluate food quality, which may lead to avoidance of food lacking certain essential nutrients or containing toxic molecules, and to alterations in developmental and metabolic programs (*Bargmann, 2006*; *Bjordal et al., 2014*; *Chi et al., 2016*; *Chng et al., 2014*; *Ha et al., 2010*; *Iwatsuki and Torii, 2012*; *Kniazeva et al., 2015*; *Melo and Ruvkun, 2012*; *Tang and Han, 2017*; *Watson et al., 2014*). Besides neuronal sensory systems that permit rapid feeding decision, food quality is also evaluated by the intestine-initiated systems that may be more sensitive in detecting the deficiency of certain types of nutrients including micronutrients and are capable of dictating changes in cellular/developmental programs as well as food uptake and seeking behaviors. However, the metabolic and signaling events in the intestine that are involved in evaluating the availability of specific nutrients, and the mechanisms underlying the connection of these signaling activities to food uptake/foraging behaviors, as well as developmental programs, remain largely unexplored. More specifically, although the functions of TOR complexes in responding to cellular nutrient changes (e.g. amino acids and ATP) have been extensively studied (*Chin et al., 2014*; *Dennis et al., 2001*; *Efeyan et al., 2015*; *Zhu et al., 2013*), the functions of these sensing activities in whole animals under true physiological conditions, including responses to nutrient variations in food, remain to be investigated.

Vitamin $B_2$ (VB2) is the precursor and component of flavin mononucleotide (FMN) and flavin adenine dinucleotide (FAD) that are the redox cofactors of a large number of flavoproteins involved in various metabolic pathways (*Joosten and van Berkel, 2007*; *Lienhart et al., 2013*; *Powers, 2003*). Animals obtain VB2 from diet and likely also from gut microbes, although the VB2 contribution from

**eLife digest** Animals are able to sense changes in the quality of their diet, which allows them to adapt to environments where the availability of nutrients fluctuates. For example, if a food lacks essential nutrients, or contains toxic molecules, animals can avoid it. If food is scarce, they can alter their metabolism to compensate. This assessment is done in the intestine, but exactly how it works is not fully understood, especially when it comes to vitamins.

Animals require Vitamin B2 to grow and remain healthy. This vitamin is involved in the process by which cells make chemical energy, which is needed to fuel many biological processes. Vitamin B2 is found in the diet, and is also produced by bacteria living in the gut. Here, Qi et al. used a worm called *Caenorhabiditis elegans* to examine how the gut detects this vitamin, and what impact this has on how the worm behaves and develops.

The experiments show that when the worms' diet includes live bacteria, they developed normally from larvae into adults. However, if the worms were fed only bacteria that had been killed by cooking and exposure to light – which damages Vitamin B2 – they stopped eating, shut down the production of two key digestive enzymes and stopped growing. Supplying these worms with extra vitamin B2, or extra digestive enzymes, stimulated the worms to start eating the cooked bacteria.

Further experiments show that when Vitamin B2 is scarce, the levels of chemical energy in the worms' cells decrease. This drop in energy is sensed by a complex of molecules called TORC1, which triggers the changes in the worms' metabolism and behavior.

The findings of Qi et al. indicate that gut microbes can play important roles in providing micronutrients like Vitamin B2. Many of the molecular pathways used by these worms have equivalents in humans. Therefore, further research on these pathways in worms may help us to understand how the human body responds to nutrients and how metabolic diseases may alter these pathways.

gut microbes has not been well documented. VB2 deficiency has been associated with various human diseases and health problems (*Powers, 2003*). It would thus be reasonable to speculate that a food-quality monitoring system in animals can sense VB2 in food and then regulate food response behaviors, and such a monitoring system may function in the intestine. Such a potential VB2-sensing mechanism is explored in this study.

## Results

### Heat-killed bacteria are low-quality food that *C. elegans* cannot use

When fed live *E. coli* strain OP50, a standard laboratory food for *C. elegans*, all hatched larvae develop to adults within 3 days at 20°C. However, when worms were fed OP50 that was killed by heat treatment (75°C for 90 min; HK-OP50 hereafter), they arrested development at early larval stages (L1-L2) and failed to consume the bacterial lawn (*Figure 1A*), suggesting that the heat-killed bacteria lacked certain nutrients or molecules required for larval growth, which is consistent with published observation suggesting that *E. coli* contains heat-labile nutrients required for *C.elegans* normal growth and longevity (*Lenaerts et al., 2008*). We performed two behavior assays to assess food dwelling and food choice (*Brandt and Ringstad, 2015*; *Fujiwara et al., 2002*; *Kniazeva et al., 2015*; *Melo and Ruvkun, 2012*; *Shtonda and Avery, 2006*). The results demonstrated a strong discrimination against HK-OP50 by wild-type worms (*Figure 1B and C*). Interestingly, in the food choice assay, the live OP50 lawn was favored until it was consumed at day 4, when most worms then moved to the HK-OP50 and consumed that lawn (*Figure 1C*), suggesting that worms eat heat-killed bacteria if they have first obtained some benefit from live bacteria. Moreover, when live-OP50 was added to worms grown on HK-OP50 plate for 30 days, they recovered to adults with viable progeny (*Figure 1—figure supplement 1A*), suggesting nutrient deficiency in HK-OP50 induced a protective response from the worms similar to starvation response.

To further investigate the impact of this interaction with live bacteria, we employed *Staphylococcus saprophyticus* (SS), a bacterium species that is inedible for *C. elegans* (*Figure 1D*). When eggs

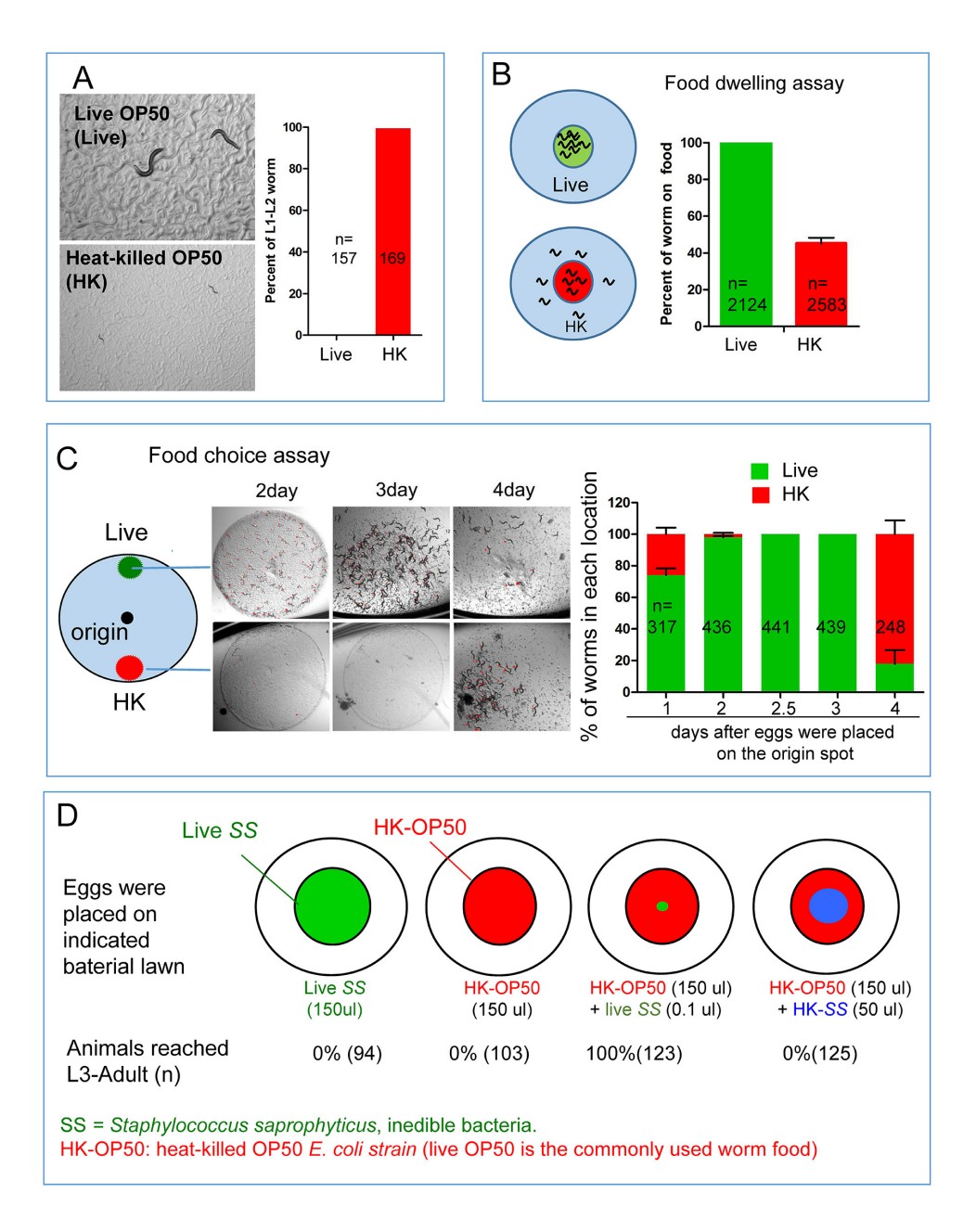

**Figure 1.** *C. elegans* do not eat heat-killed bacteria without interacting with live bacteria. (**A**) Microscope images and bar graph showing that worms fed heat-killed OP50 (HK-OP50 or HK) arrested at L1-L2 stage three days after hatching. (**B**) Schematic drawing and quantitative data of the food dwelling assay. Circles indicate the food spot for live (green) and HK-OP50 (red) bacteria, respectively. The animals were scored 24 hr after L1 worms were placed on the food spot. Data are represented as mean ±SD. (**C**) Schematic drawing, microscopic images and quantitative data of the food choice assay. Eggs were place in the center spot (origin). Live (green) and heat-killed (red) bacteria were placed on opposite sides of the plate. The percentage of worms on each spot was calculated at the indicated time. Worms moved to heat-killed bacteria and consumed it after 4 days when live food was totally consumed. Data are represented as mean ±SEM. (**D**) An assay for the effect of a small amount of live, non-edible bacteria on the growth of animals fed heat-killed bacteria. The colored diagrams show the feeding conditions. Percentage of animals that grew to L3-adult stages is indicated below each diagram. The combination of HK-OP50 and small amount live SS, neither of which alone can support growth, could support food uptake and growth. The control (4th column) suggests that only live SS can promote worms to consume HK-OP50. For bar graphs, number of worms scored is indicated in each bar. All data are representative of at least three independent experiments.

The following source data and figure supplement are available for figure 1:

*Figure 1 continued on next page*

*Figure 1 continued*

**Source data 1.** Numerical data of *Figure 1A–1D* and *Figure 1—figure supplement 1C*.
**Figure supplement 1.** Impact of interaction with live bacteria on the ability of *C. elegans* to consume heat-killed bacteria.

were placed on HK-OP50 lawn containing a tiny amount of live *SS*, the hatched worms were able to consume HK-OP50 and grow, suggesting that the live bacteria provided micronutrients that rendered the heat-killed bacteria edible (*Figure 1D* and *Figure 1—figure supplement 1B*). Additional tests indicated that such potential components did not reach the worms through odorants and suggested that worms likely obtain the benefit from the small amount live bacteria through ingestion (*Figure 1—figure supplement 1C*; *Figure 1C*). These data led to a speculative model that the trace amount of live bacteria, ingested by the worms, stayed in the intestine and continuously secreted micronutrients into the gut tract, which is similar to microbes in mammalian gut.

## Vitamin B$_2$ supplementation promotes the usage of heat-killed bacteria as food

Deficiency of certain nutrients in HK-OP50 may be detected by a potential food quality evaluation mechanism leading to changes in food uptake and foraging behavior. Among micronutrients tested, we found that vitamin B2 (VB2) supplementation partially but significantly recovered larval growth as indicated by increased gonad length at day 7, which indicated the increased usage of the food (*Figure 2A* and *Figure 2—figure supplement 1A*). These animals fed VB2 supplemented food still grew very slowly, reaching mid to late larval stages based on gonad length and vulval development compared to that of well-fed worms, but never reach adulthood by day 7 (*Figure 2—figure supplement 1B,C*). This slow growth indicated that VB2 supplementation does not completely compensate for the undefined deficiencies of heat-killed bacteria, which suggested that multiple essential nutrients are lacking in heat-killed bacteria. However, the prominent increase in growth suggested that VB2 may play a critical role in a food-quality evaluation pathway. Our behavior assays showed that VB2 supplementation increased the preference for heat-killed bacteria (*Figure 2B* and *Figure 2—figure supplement 2A*). Using HPLC-UV (*Howe et al., 2015*), we found the VB2 level was drastically reduced in worms fed HK-OP50, compared to worms fed live OP50, and the level was mostly recovered by VB2 supplementation (*Figure 2C*), suggesting VB2 was very low in heat-killed bacteria. To confirm this, we measured the VB2 level in bacteria and found that VB2 level is low in heat-killed OP50 (*Figure 2—figure supplement 2C*).

## Vitamin B2 promotes intestinal protease activity

To understand how VB2 affects food uptake in worms, we tested the role of the digestive system by feeding worms with HK-OP50 culture pre-treated with exogenous digestive enzymes. We found that larval growth was enhanced only by protease-treated HK-OP50 (*Figure 2D*), indicating that exogenous protein digestion improved usability of the heat-killed bacteria. We then observed that HK-OP50 supplemented with a protease inhibitor cocktail (PIC) eliminated the beneficial effect of VB2 supplementation (*Figure 2E*). These results suggest that worms fed HK-OP50 have reduced protease activity, and that the benefit of VB2 supplementation is dependent on endogenous proteases.

We then used a protease detection assay (*Hama et al., 2009*; *Semova et al., 2012*) to directly examine the in vivo protease activity in the intestinal tract (*Figure 2—figure supplement 2D and E*). We found that the protease activity was drastically reduced in worms fed HK-OP50, compared to worms fed live OP50, and VB2 supplementation partially recovered the protease activity (*Figure 2F*). Taken together, these data support that worms down-regulate digestive enzymes in response to VB2 deficiency.

## Proteases ASP-13 and ASP-14 and GATA factor ELT-2 play critical roles in VB2-promoted protease activity and food usability

To identify the proteases regulated by VB2, we examined the mRNA levels of 27 intestine-enriched, protease genes (*McGhee et al., 2007*). qRT-PCR analysis showed that the mRNA levels of four

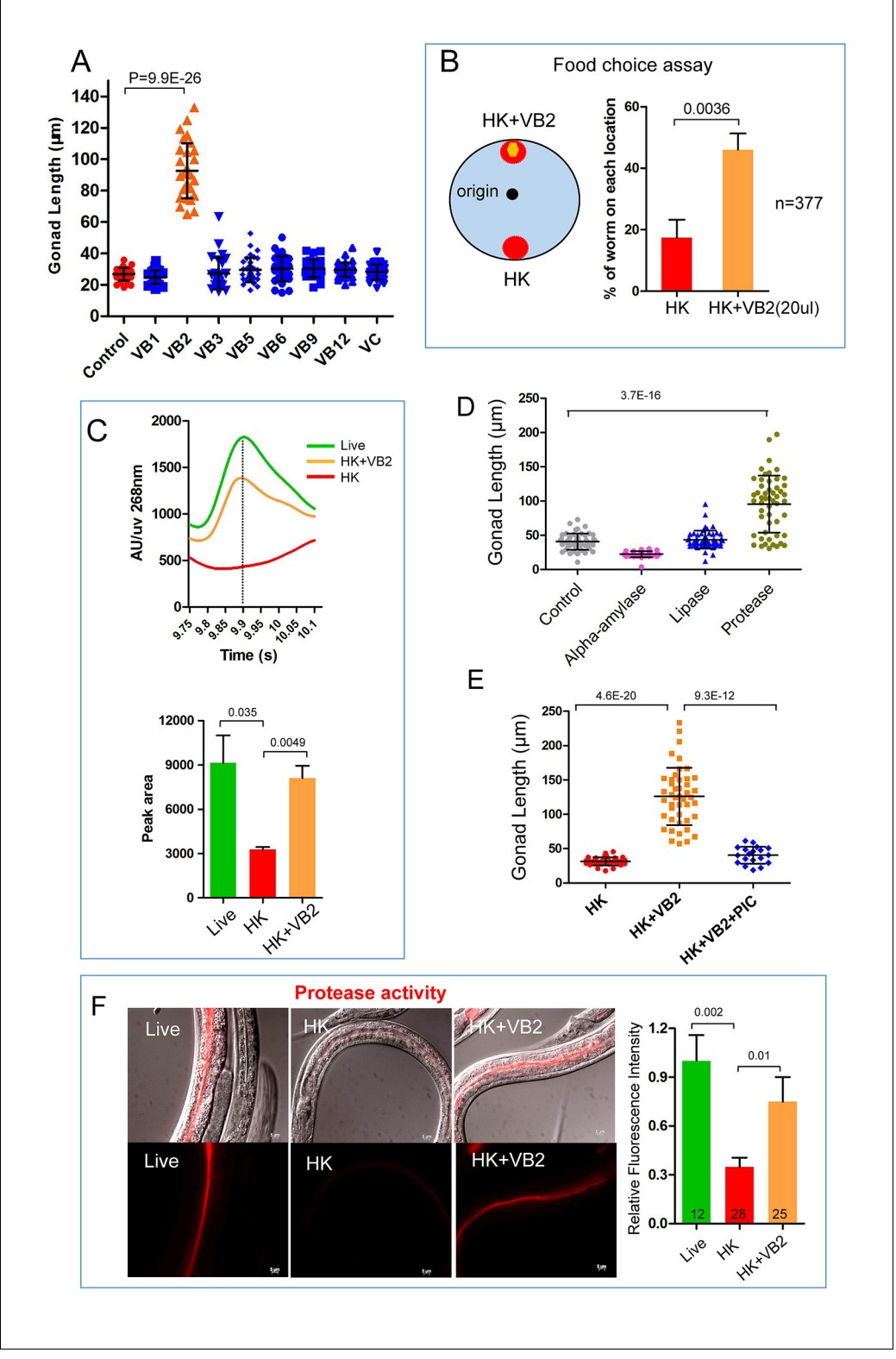

**Figure 2.** Vitamin B2 (VB2) supplementation increases the usage of heat-killed bacteria and promotes intestinal protease activity. (**A**) Scatter plot of gonad length of worms fed heat-killed OP50 supplemented with indicated vitamins scored at Day 7. > 28 worms were scored for each sample. Representative images are shown in *Figure 2—figure supplement 1A*. Data are represented as mean ±SD. (**B**) Food-choice assay to test the effect of VB2 supplementation on animal dwelling behavior. The worms were scored for the ratio between the two

*Figure 2 continued on next page*

*Figure 2 continued*

locations at 3 day. Data are represented as mean ±SD. (**C**) HPLC-UV analysis of VB2 extracted from worms fed live, HK-OP50 and HK-OP50+VB2 supplementation. The bar graph represents the area counts of VB2 UV absorption peaks (identified by Analyst software), the mean ±SEM of three independent experiments. A graph for VB2 standard from HPLC-UV analysis is shown in *Figure 2—figure supplement 2B*. (**D**) Scatter plot showing the effect of treating HK-OP50 with indicated enzymes, on larval growth (by measuring gonad length). Only protease treatment significantly promoted growth. The number of worms scored was 52, 37, 52 and 52, respectively. Data are represented as mean ±SD. (**E**) Scatter plot showing the effect of a protease inhibitor cocktail (PIC) on the growth of larvae fed HK-OP50 supplemented with VB2. The number of worms scored was 32, 44 and 18, respectively. Data are represented as mean ±SD. (**F**) Fluorescence images and bar graph showing the impact of heat-killed bacteria and VB2 supplementation on in vivo protease activity in the intestinal tract by the EnzChek protease assay (*Hama et al., 2009*). L2 stage worms were incubated with quenched BODIPY TR-X casein for 3 hr before imaging. Data are represented as mean ±SEM. p-Values were calculated by T-test and p<0.05 was considered a significant difference. For bar graphs, number of worms scored is indicated in each bar. All data are representative of at least three independent experiments.

The following source data and figure supplements are available for figure 2:

**Source data 1.** Numerical data of *Figure 2A–2F* and *Figure 2—figure supplement 2A, D and E*.

**Figure supplement 1.** Gonad length of worms under different conditions.

**Figure supplement 2.** Impact of vitamin B2 supplementation on the ability of *C. elegans* to consume heat-killed bacteria.

---

genes, *asp-13*, *asp-14*, *cpr-1* and *cpr-4,* were dramatically increased in worms fed HK-OP50+VB2 compared to that in worms fed HK-OP50 (*Figure 3A*). Among these four genes, *asp-13* and *asp-14* were also highly expressed in worms fed on live bacteria (*Figure 3A*). We then found that knocking down both *asp-13* and *asp-14* resulted in a strong suppression of the effect of VB2 supplementation (*Figure 3B*), suggesting that these enzymes are essential for the VB2 effect. Over-expression of both *asp-13* and *asp-14* behind a ubiquitously expressed *rpl-28* promoter (termed *asp-13/14* OE) (*Zhang et al., 2009*) significantly improved the food uptake and dwelling behavior of worms fed HK-OP50, as determined by the gonad length, food choice and seeking assays (*Figure 3C–E*). More-over, intestinal-specific RNAi of *asp-13/14* prevents the rescue effect of VB2 supplementation on worms fed HK-food (*Figure 3—figure supplement 1A and B*), suggesting that intestinal function of ASP-13 and ASP14 are essential for the VB2 effect. This is consistent with the previous finding that *asp-13/14* is highly expressed in the intestine (*McGhee et al., 2007*) and the results from our in vitro protease/protease inhibitor assay (*Figure 2D*). These data strongly support a critical role of ASP-13 and ASP-14 in the VB2-induced increase in protease activity and uptake of heat-killed bacteria.

To identify transcription factors that regulate the expression of intestinal protease genes in response to VB2 supplementation, we combined RNAi with the protease assay to screen 18 tran-scription factors previously shown to be enriched in the intestine (*McGhee et al., 2007*) (*Figure 3—figure supplement 2A*). The protease activity was most dramatically decreased in worms treated with *elt-2(RNAi)* (*Figure 3F* and *Figure 3—figure supplement 2A*), which is consistent with the recent report that *elt-2(RNAi)* decreases the expression of ~50% of known protease genes, including *asp-13* and *asp-14* (*Mann et al., 2016*). Using an integrated, translational ELT-2::GFP reporter strain which is in a *glo-4* mutant background to reduce auto fluorescence (*Mann et al., 2016*), we found that the ELT-2::GFP signal was decreased in worms fed HK-OP50 and the reduction was mostly recovered by VB2 supplementation (*Figure 3G*). We also found that decreased ELT-2 on heat-killed food was recovered by VB2 supplementation in wild-type worm (*Figure 3G*, Western blot), support-ing that ELT-2 expression responds to changes in nutrient status. Using an established ELT-2 over-expression [*elt-2 (OE)*] strain (*Mann et al., 2016*), we found that over-expression of ELT-2 signifi-cantly increased both larval growth measured by gonad length and protease activity in worms fed HK-OP50, eliminated the preference for HK-OP50+VB2 over HK-OP50 in the food choice assay, and increased the expression of *asp-13* and *asp-14* (*Figure 3H and I*; *Figure 3—figure supplement 2B and C*). Therefore, increased expression of *elt-2* is both necessary and sufficient for the impact of

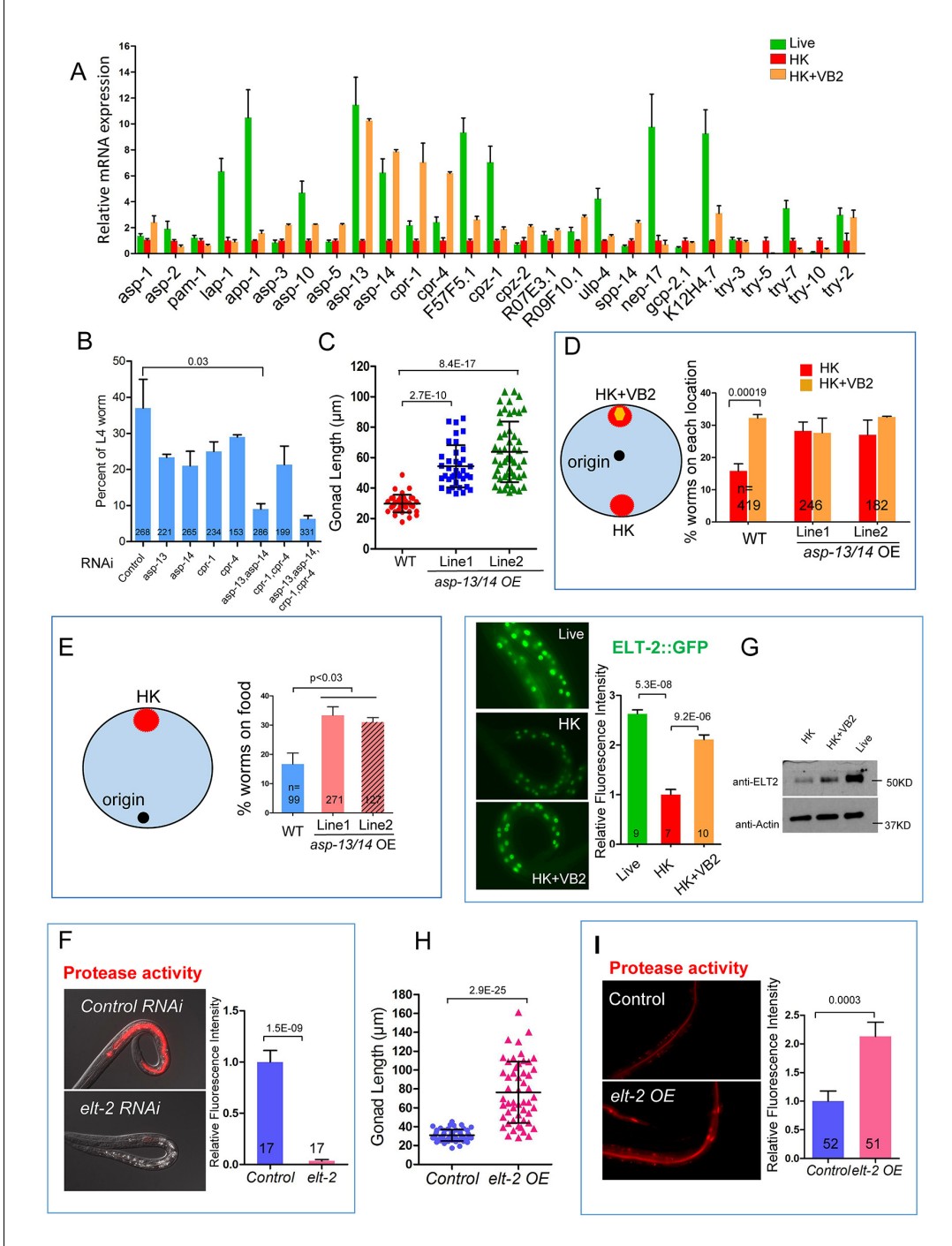

**Figure 3.** Proteases ASP-13 and ASP-14 and GATA factor ELT-2 play critical roles in VB2-promoted food usability and animal growth. (**A**) Results of qRT-PCR analysis showing the expression of indicated protease genes in wild-type worms fed indicated food. Data are represented as mean ±SEM. (**B**) Impact of RNAi of indicated genes on larval growth when fed HK-OP50+VB2 supplementation. Worms with both *asp-13* and *asp-14* knocked down displayed strongest effect. Data are represented as mean ±SEM. (**C**) Scatter plot showing the impact of over-expression of the two protease genes behind the *rpl-28* promoter, measured by gonad length of larvae fed HK-OP50 at Day 7. n = 41, 39 and 52, respectively. Data are represented as mean ±SD. (**D and E**) Cartoon diagram and data from food choice (**D**) and food seeking (**E**) assays showing that overexpression of *asp-13* and *asp-14* (*asp-13/14* OE) eliminated the discrimination against HK-OP50 over HK-OP50+VB2 (**D**) and improved affinity of worms towards HK-OP50 (**E**). Data from two different overexpression lines are shown. Data are represented as mean ±SEM. (**F**) Fluorescence images and bar graph showing protease activity in the intestinal tract is dramatically decreased in worms treated with *elt-2(RNAi)*. Data are represented as mean ±SEM. (**G**) Fluorescence images (integrated translational ELT-2::GFP reporter strain) and Western blot (wild-type strain) showing that the ELT-2 expression is prominently decreased in worms fed

*Figure 3 continued on next page*

*Figure 3 continued*

HK-OP50 compared to that in worms fed live OP50, and the expression is largely recovered by VB2 supplementation. Data are represented as mean ±SEM. (**H**) Scatter plot showing larval growth measured by gonad length of worms fed heat-killed OP50 is increased by an *elt-2* over-expression transgene (OE) at Day 7. n = 52 for each condition. Data are represented as mean ±SD. (**I**) Fluorescence images and bar graph showing that the protease activity is increased in *elt-2* overexpression (*elt-2 OE*) worms fed HK-OP50. p-Values were calculated by T-test. For bar graphs, number of worms scored is indicated in each bar. Data are represented as mean ±SEM. p-Values were calculated by T-test and p<0.05 was considered a significant difference. For bar graphs, number of worms scored is indicated in each bar. All data are representative of at least three independent experiments.

The following source data and figure supplements are available for figure 3:

**Source data 1.** Numerical data of *Figures 3B–2I* and *Figure 3—figure supplements 1A, B*, *2A, C and E*.
**Figure supplement 1.** VB2 supplementation-induced increase in protease activity and growth rescue in worms fed HK-food depend on intestinal expression of ASP-13 and ASP-14.
**Figure supplement 2.** Roles of protease ASP-13 and ASP-14, and GATA factor ELT-2 in VB2-induced food uptake and growth of worms fed heat-killed bacteria.

VB2 on protease expression and food usability. Additional tests supported the hypothesis that down-regulating *asp-13* and *asp-14* by a low ELT-2 level critically contributes to the poor food usage in worms fed HK-OP50 (*Figure 3—figure supplement 2D and E*).

## TORC1 mediates VB2-dependent regulation of protease activities and worm growth

Despite extensive studies on nutrient sensing by TORC1(*Efeyan et al., 2015*), its role in food quality evaluation processes, which include not only the detection of deficiency of a specific nutrient, but also specific downstream physiological responses in live animals, remain to be explored, particularly for micronutrients such as VB2. We first tested if TORC1 is required for VB2-induced activation of digestive enzymes in worms fed heat-killed bacteria. We found that pretreatment with RNAi targeting either *daf-15/raptor* or *ragc-1*, encoding two key TORC1 components (*Jia et al., 2004*; *Long et al., 2002*; *Robida-Stubbs et al., 2012*), essentially eliminated the effect of VB2 supplementation on protease activity (*Figure 4A*), indicating that VB2 induction of protease activity is TORC1 dependent. To address that lower TORC1 activity on protease activity derive does not simply from slowing or arresting development, we tested the protease activity of several arrested larvae (RNAi treatment). *Figure 4—figure supplement 1* shows that protease activity is not all reduced in arrested larvae, which indicate that lower TORC1 activity on protease activity (*Figure 4A*) is unlikely to be from generally slowing or arresting development.

We then measured TORC1 activity by utilizing an established autophagy marker (LGG-1::GFP puncta) known to be repressed by TORC1 (*Robida-Stubbs et al., 2012*). We found that worms fed HK-OP50 displayed a high level of LGG-1::GFP puncta in seam cells compared to that in worms fed live OP50, and this increase was suppressed by VB2 supplementation (*Figure 4B*). To support that the observed change in the LGG-1::GFP puncta level under our testing conditions reflects a specific change in TORC1 activity, we showed that the elevated LGG-1::GFP puncta level in worms fed HK-OP50 was suppressed by hyperactivation of TORC1 in a *nprl-3* loss-of-function (*lf*) mutant (*Figure 4—figure supplement 2A*). NPRL-3 is a negative regulator of TORC1 in both worms and mammalian cells (*Bar-Peled et al., 2013*; *Zhu et al., 2013*). Conversely, inhibition of TORC1 activity by *daf-15* (*RNAi*) elevated the LGG-1::GFP puncta level in well-fed, young larvae (*Figure 4—figure supplement 2B*). In addition, apical localization of VHA-6, which we previously showed contributes to intestinal TORC1 activation by a lipid biosynthesis pathway (*Zhu et al., 2015*), was altered in worms fed HK-OP50 and this defect was suppressed by VB2 supplementation (*Figure 4—figure supplement 2C and D*). These data indicate that TORC1 activity is decreased in worms fed heat-killed bacteria and that VB2 supplementation significantly recovers the activity.

We obtained further evidence that TORC1 acts downstream of VB2 by showing that TORC1 activation is sufficient to mimic the effect of VB2 in boosting intestinal protease activity and food usage.

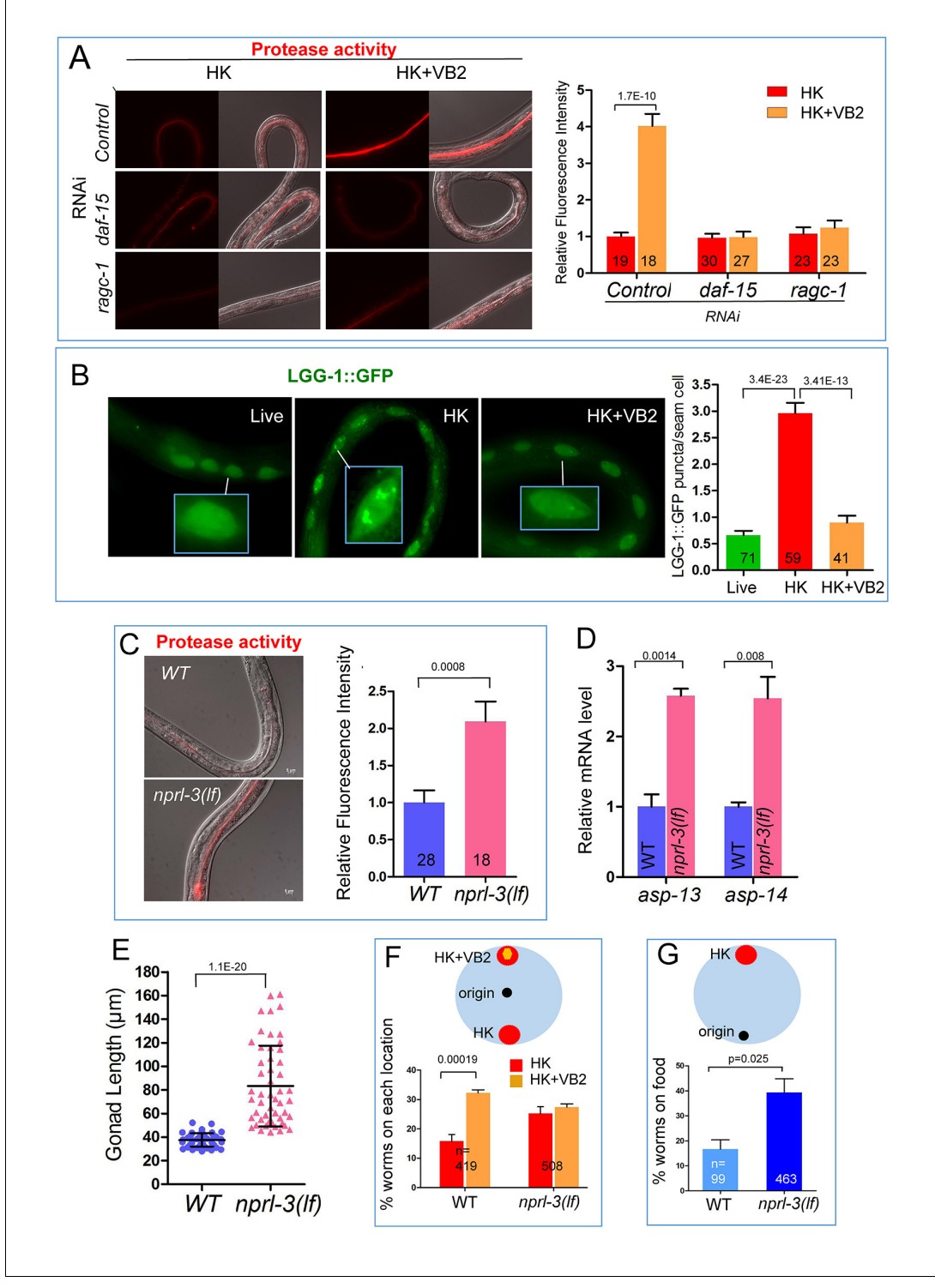

**Figure 4.** TORC1 mediates the impact of VB2 on the expression of digest enzymes and food usage. (**A**) Fluorescence images and bar graph showing that VB2-induced increase in protease activity in the digestive tract depends on TORC1 function. Data are represented as mean ±SEM. (**B**) Fluorescence images and bar graph showing that autophagy activity (measured by the intensity of LGG-1::GFP puncta in seam cells) increases in worms fed HK-OP50, compared to worms fed live OP50, and the increase is eliminated by VB2 supplementation. The inset image in each panel shows magnified area indicated by the white bar. *Figure 4—figure supplement 2A and B* show data supporting that LGG-1 puncta is repressed by TORC1 activity, consistent with known negative regulation of autophagy by TORC1 (*Robida-Stubbs et al., 2012*). Data are represented as mean ±SEM. (**C**) *Figure 4 continued on next page*

*Figure 4 continued*

Fluorescence images and bar graph showing that protease activity is increased in an *nprl-3(lf)* mutant fed HK-OP50. *nprl-3* negatively regulates TORC1 (*Zhu et al., 2013*). Data are represented as mean ±SEM. (D) qRT-PCR analysis showing that mRNA of *asp-13* and *asp-14* were increased in *nprl-3(lf)* mutant fed HK-OP50. Data are represented as mean ±SEM. (E) Scatter plot showing that larval growth is increased in the *nprl-3(lf)* mutant worms fed HK-OP50 at Day 7. n = 52 for each condition. Data are represented as mean ±SD. (F and G) Result from food choice (F) and food-seeking (G) assays showing that *nprl-3(lf)*, which hyperactivates TORC1, eliminated the discrimination against HK-OP50 over HK-OP50+VB2 (F) and improved affinity of worms toward HK-OP50 (G). The wild-type data are the same as that in *Figure 3D and E*, as the data for both pairs of figures were generated from the same set of experiments. Data are represented as mean ±SEM. p-Values were calculated by T-test and p<0.05 was considered a significant difference. For bar graphs, number of worms scored is indicated in each bar. All data are representative of at least three independent experiments.

The following source data and figure supplements are available for figure 4:

**Source data 1.** Numerical data of *Figure 4A–4C and E–G* and *Figure 4—figure supplements 1*, *2A, B, D, F and G*.
**Figure supplement 1.** Protease activity of several arrested larvae (RNAi treatment), *elt-2* RNAi as positive control.
**Figure supplement 2.** LGG-1::GFP as a marker for activity downstream of TORC1 and the impact of food source on apical membrane polarity and VHA-6 localization in the intestine of *C. elegans.*

Specifically, we found that the *nprl-3(lf)* mutation significantly improved intestinal protease activity, the expression of *asp-13* and *asp-14*, larval growth measured by gonad length and dwelling behavior of worms fed HK-OP50 (*Figure 4C–G*); however, this rescue was not because VB2 level was recovered as VB2 level in *nprl-3 (lf)* mutants fed heat-killed food did not increase (*Figure 4—figure supplement 2E*). In contrast, reducing TORC1 activity (*daf-15 RNAi*) led to defects in food choice (*Figure 4—figure supplement 2F*). Moreover, *daf-15(RNAi)* treatment decreased the expression of the translational ELT-2::GFP reporter in the reporter strain and the endogenous ELT-2 protein in the wild-type (*Figure 4—figure supplement 2G*), which is consistent with the observation in a previous report (*Schieber and Chandel, 2014*). Therefore, our data strongly support that TORC1 mediates the effect of VB2 on the expression of proteases and the usage of heat-killed bacteria.

## VB2 promotes TORC1 and protease activities through FAD and ATP production

Vitamin B2, also known as riboflavin, is the precursor of flavin mononucleotide (FMN) and flavin adenine dinucleotide (FAD) (*Joosten and van Berkel, 2007*; *Lienhart et al., 2013*; *Powers, 2003*) (*Figure 5A*). Using HPLC-UV analysis, we found that both FMN and FAD levels were decreased in worms fed HK-OP50 compared to worms fed live OP50, and their levels were partially restored by VB2 supplementation (*Figure 5B*). Therefore, lack of sufficient FAD and FMN likely contributes to the low quality of heat-killed bacteria as a food source.

To test if FAD also mediates the VB2 impact on TORC1 and protease production, we showed that FAD supplementation improved food uptake, intestinal protease activity and ELT-2 expression, as VB2 does (*Figure 5C and D*; *Figure 5—figure supplement 1B*). Furthermore, the FAD benefit is also TORC1 dependent, as pretreatment with RNAi targeting *daf-15/raptor* essentially eliminated the effect of FAD supplementation on protease activity (*Figure 5D*).

We then further analyzed the functional relationship between FAD and TORC1 by decreasing FAD production through RNAi of a FAD synthetase gene (*flad-1*) and a riboflavin transporter gene (*rft-2*) (*Biswas et al., 2013*; *Gandhimathi et al., 2015*; *Liuzzi et al., 2012*) (*Figure 5A*). Protease activity was dramatically reduced in both RNAi-treated strains and the activity was recovered in *nprl-3(lf)* mutants (*Figure 5E,F,H and I*). Furthermore, both *flad-1(RNAi)* and *rft-2(RNAi)* significantly increased autophagy (*Figure 5G and J*). These data support that TORC1 acts downstream of FAD to promote protease activity. The similarity of the data between *flad-1* and *rft-2* further supports that FAD plays the key role in mediating VB2 effect on intestinal protease activity.

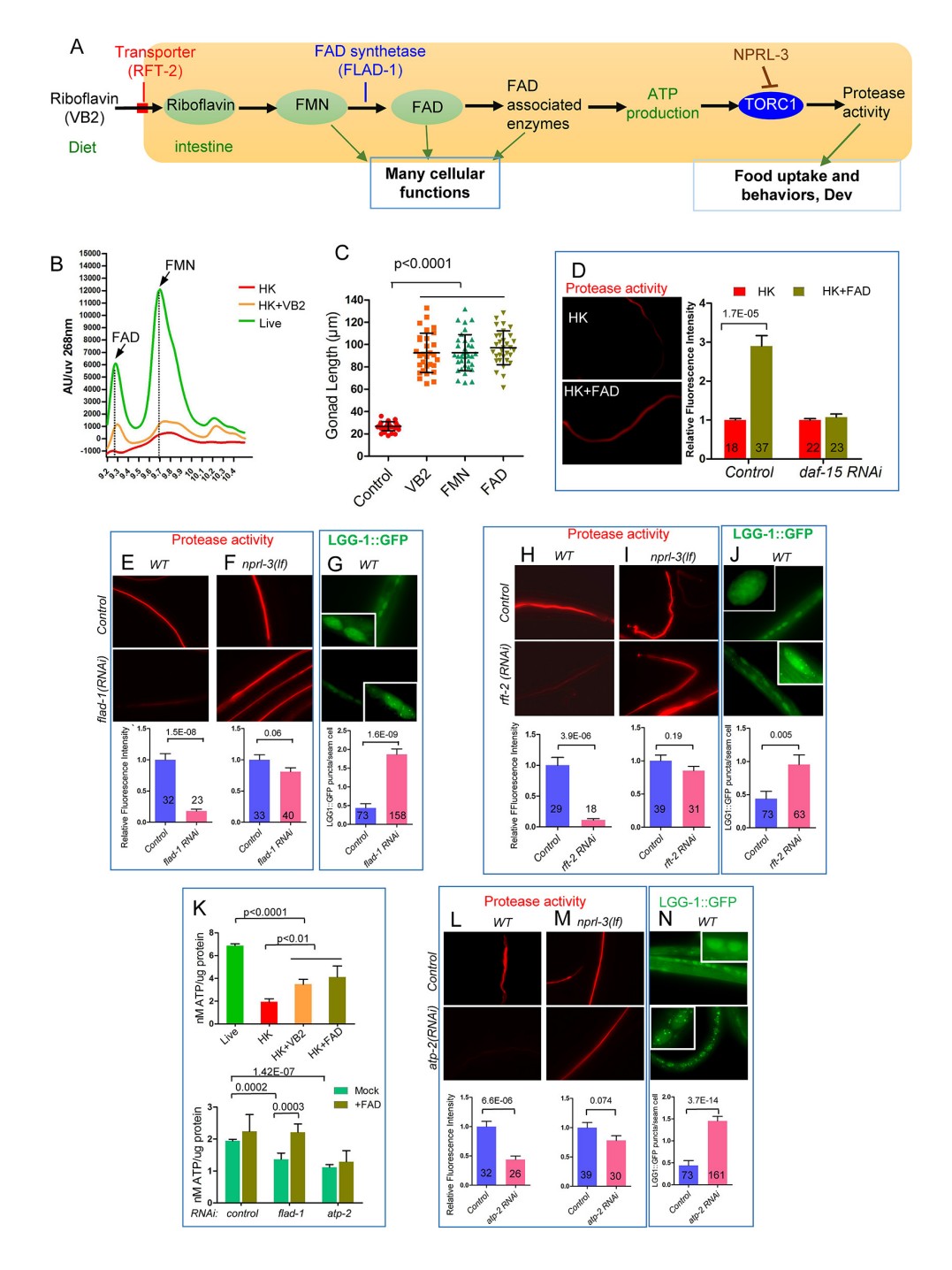

**Figure 5.** Change in FAD may mediate the effect of VB2 on food usage by worms fed heat-killed bacteria. (**A**) Cartoon diagram to illustrate known functional relationship between VB2 (riboflavin), flavin mononucleotide (FMN) and flavin adenine dinucleotide (FAD), as well as a proposed VB2 sensing pathway in the intestine, based on this study. *RFT-2, expressed in the intestine in C. elegans,* is a homolog of the riboflavin transporter three that is required for VB2 uptake (*Biswas et al., 2013*; *Gandhimathi et al., 2015*). *FLAD-1* is a FAD synthase that catalyzes the last step of FAD biosynthesis (*Liuzzi et al., 2012*). NPRL-3 represses TORC1 activity (*Zhu et al., 2013*). (**B**) HPLC-UV detection peak of FAD and FMN from worms fed under indicated conditions. *Figure 5—figure supplement 1A* shows the graph of FAD and FMN standards. (**C**) Scatter plot showing that supplementation of FMN or FAD significantly recovered the of wild-type larvae fed HK-OP50. The number of worms scored was 28, 29, 35 and 31, respectively. Data are represented as mean ±SD. (**D**) Fluorescence images and bar graph showing that FAD supplementation elevated the in vivo protease activity in worms fed HK-OP50 and the increase is eliminated by RNAi knockdown of *daf-15/raptor*. Data are represented as mean ±SEM. (**E and F**) Fluorescence images and bar graph showing that RNAi knock down of *flad-1*/FAD synthetase sharply reduced the in vivo protease activity in wild type (**E**). The reduction was

*Figure 5 continued on next page*

*Figure 5 continued*

suppressed by a *nprl-3(lf)* mutation (**F**) that hyperactivates TORC1. Data are represented as mean ±SEM. (**G**) RNAi knockdown of *flad-1* elevated autophagy activity (LGG-1::GFP puncta) that is known to be repressed by TORC1. Data are represented as mean ±SEM. (**H and I**) Fluorescence images and bar graph showing that RNAi knock down of *rft-2*, encoding a riboflavin transporter (see A), sharply reduced the in vivo protease activity in wild type. The sharp reduction was suppressed by a *nprl-3(lf)* mutation that hyperactivates TORC1. Data are represented as mean ±SEM. (**J**) RNAi reduction of *rft-2* elevated the number of LGG-1::GFP puncta that marks the level of autophagy. The increase in autophagy is consistent with a reduction in TORC1 activity. Data are represented as mean ±SEM. (**K**) Measurement of ATP levels in worms. (top) ATP level is reduced in worms fed HK-OP50 and the reduction is partially suppressed by VB2 or FAD supplementation. (bottom) *flad-1(RNAi)* or *apt-2(RNAi)* caused reduction in ATP level in wild type, and the decrease by *flad-1(RNAi)*, but not by *atp-2(RNAi)*, was overcome by FAD supplement. Data are represented as mean ±SD. (**L and M**) Fluorescence images and bar graph showing that RNAi knock down of *atp-2*/ATP synthetase, significantly reduced the intestinal protease activity in wild type (**L**). The reduction seen was suppressed by an *nprl-3(lf)* mutation (**M**). Data are represented as mean ±SEM. (**N**) RNAi knock down of *atp-2* elevated the autophagy activity (LGG-1::GFP puncta). Data are represented as mean ±SEM. p-Values were calculated by T-test and p<0.05 was considered a significant difference. For bar graphs, the number of worms scored is indicated within each bar. All data are representative of at least three independent experiments.

The following source data and figure supplement are available for figure 5:

**Source data 1.** Numerical data of *Figure 5B–5N* and *Figure 5—figure supplement 1B*.
**Figure supplement 1.** Measurement of FAD and FMN standards and role of FAD supplementation on ELT-2 expression.

FAD-associated proteins are involved in many metabolic events including fatty acid metabolism, the TCA cycle and electron transport chain (*Lienhart et al., 2013*). Since many of these events are related to energy production, and TORC1 is known to be an ATP sensor in mammals and *C. elegans* (*Chin et al., 2014*; *Dennis et al., 2001*; *Hay and Sonenberg, 2004*), we tested if the ATP level serves as a signal to relay the impact of VB2/FAD on TORC1 and downstream protease activity. We found that the ATP level was indeed dramatically reduced in worms fed HK-OP50 compared to that in worms fed live OP50, whereas VB2/FAD supplementation partially recovered the level (*Figure 5K*). Consistently, reducing FAD production by *flad-1(RNAi)* in the first generation of wild-type worms also reduced ATP and the reduction was eliminated by FAD supplementation (*Figure 5K*). Therefore, the decreased ATP level is a major cellular consequence of VB2 deficiency. To provide further evidence that ATP reduction is causal to the reduction in TORC1 and protease activity, we used RNAi to partially inactivate *atp-2/ATP5B* that encodes an ATP synthetase known to affect ATP production in worms (*Chin et al., 2014*). We also observed a modest reduction of ATP level in the first generation of *atp-2(RNAi)*-treated animals and the reduction was not overcome by FAD supplementation (*Figure 5K*), consistent with *atp-2* acting downstream of FAD regarding ATP production. Importantly, intestinal protease activity was *prominently decreased in the second generation of atp-2(RNAi)-treated animals and the reduction was partially suppressed by nprl-3(lf)* (*Figure 5L and M*). Furthermore, LGG-1::GFP puncta were increased in *atp-2(RNAi)* worms fed live OP50 (*Figure 5N*), which is consistent with a previous observation in *C. elegans* (*Chin et al., 2014*). Together, these data support the hypothesis that the ATP level change mediates the impact of VB2 on TORC1 and intestinal protease activity. Our data do not exclude potential impacts of other metabolic products from FAD-associated enzymes in the process.

## Discussion

We identified an intestinal food-quality evaluation mechanism by which animals detect vitamin B2 deficiency by the FAD-ATP-TORC1-ELT-2 pathway that dictates food uptake and foraging behaviors (*Figure 5A*). Such a mechanism with conserved functions enhances survival in wild environments by discriminating against low-quality food. Unlike rapid neuronal food responses that facilitate quick food consumption decisions, the intestinal system described here requires a rather slow, multi-step process after the ingestion of food. It should be noted that our findings do not exclude the involvement of sensory neurons in the food evaluation systems described in this study. The intestinal process may be evolved specifically to detect the deficiency of specific nutrients such as vitamins that elude the neuronal sensory system that may be honed to detect the presence of major nutrients and toxins (*Bargmann, 1997*; *Chandrashekar et al., 2006*; *Watson et al., 2015*). In addition to VB2, we

have recently shown that intestinal deficiencies in certain fatty acids and pyrimidine are also sensed by signaling systems in *C. elegans* to alter reproductive/developmental programs (*Chi et al., 2016*; *Kniazeva et al., 2015*). These examples also suggest that the protective response to nutrient deficiency by the intestine-initiated mechanism commonly includes regulation of developmental/reproductive programs, in addition to food behaviors. VB2 deficiency likely affects the functions of a large number of flavoproteins involved in many essential cellular functions; the responses by TORC1, including protease regulation and developmental arrest, may be considered the last and emergent protective actions by the animal.

TORC1 is well known for its role in sensing multiple biomolecules including amino acids, lipids and ATP (*Chin et al., 2014*; *Dennis et al., 2001*; *Efeyan et al., 2015*; *Zhu et al., 2013*). However, what this ATP-TORC1-sensing activity might do in a whole organism under physiological conditions including response to environmental challenges is a fascinating and largely unexplored question. This study provides a perfect example to address this question as we present evidence for that ATP serves as an intermediate nutrient to mediate the sensing of VB2/FAD by TORC1. Moreover, our study may be the first to connect TORC1 sensing to a specific micronutrient in a model organism. The versatility of TORC1 in responding to the deficiency of multiple nutritional/metabolic inputs suggests that the TORC1-protease pathway may be used to evaluate the levels of multiple essential nutrients in food sources, albeit that the evaluation process for other nutrients in the intestine remain to be demonstrated in vivo.

This study may also indicate the importance of live bacteria in providing micronutrients, such as VB2, to animals and may thus be highly related to human health, as we obtain vitamins from gut microbes in addition to food sources (*Albert et al., 1980*; *LeBlanc et al., 2013*). Since VB2 is known to be light sensitive and heat labile in aqueous solution (*Sheraz et al., 2014*), the contribution of VB2 from gut microbes could be significantly undervalued. The system used in this study for analyzing VB2 may potentially be effective to investigate the roles and sensing mechanisms for other micronutrients provided by microbes.

# Materials and methods

## *C. elegans* strains and maintenance

Nematode stocks were maintained on nematode growth medium (NGM) plates seeded with bacteria (E. coli OP50) at 20°C (http://www.wormbook.org/). The following strains/alleles were obtained from the Caenorhabditis Genetics Center (CGC) or as indicated: N2 Bristol (termed wild type), KWN117{*pha-1(e2123) III; him-5(e1490) V; rnyEx60 [pELA2 (vha-6p:::vha-6::mCherry) + myo-3p::GFP + pCL1 (pha-1+)]*}(RRID:WB-STRAIN:KWN117), DA2123{*adIs2122 [lgg-1p:GFP::lgg-1 + rol-6 (su1006)]*}(RRID:WB-STRAIN:DA2123), MH4672[*nprl-3(ku540)*] (*Zhu et al., 2013*), SD1949{*glo-4 (ok623);gaIs290 [elt-2::TY1::EGFP::3xFLAG(92C12) + unc-119(+)]*} (RRID:WB-STRAIN:SD1949), SD1965{*rde-1(ne300); unc-119(ed3); Ex[unc-119(+)]*}(RRID:WB-STRAIN:SD1965), SD1963{*unc-119 (ed3) III; rde-1(ne300) V;gaEx234[elt-2p::elt-2::GFP + unc-119(+)]*}(RRID:WB-STRAIN:SD1963); VJ402 (*erm-1p-erm-1::GFP*) (RRID:WB-STRAIN:VJ402) (*Göbel et al., 2004*);VP303 {*rde-1(ne219) V; [nhx-2p:: rde-1 + rol-6(su1006)]*} (RRID:WB-STRAIN:VP303).

Ex [*rpl-28p::asp-13+rpl-28p::asp-14*]; this strain was constructed for this study by standard cloning and microinjection techniques.

## Preparation and feeding of worm food with various treatments

Preparation of HK-OP50: standard overnight culture of *E. coli* OP50 grown in LB broth was concentrated to 1/10 vol and was then heat-killed in a 75°C water bath for 90 min. About 150 µl of the HK-OP50 was spread onto each NGM plate.

For vitamin supplementation, each vitamin was dissolved in H2O to generate a stock. The stock solution was spread onto NGM plates seeded with HK-OP50. The vendor, stock concentrations and volumes used for each vitamin are listed as follows: Thiamine hydrochloride VB1 (Sigma T4625, 10 mg/ml, 30 µl), (−)-Riboflavin/VB2 (Sigma R9504 0.3 mg/ml, 50 µl), Nicotinamide/VB3 (Sigma, N3376, 30 mg/ml, 30 µl), D-Pantothenic acid hemicalcium salt/VB5 (Sigma 21210–5 G-F, 30 mg/ml, 30 µl), Pyridoxine hydrochloride/VB6 (Sigma P9755, 4 mg/ml, 30 µl), Folic acid/VB9 (Sigma F-8259, 2 mg/

ml, 30 µl), VB12 (Sigma V2876, 30 ug/ml, 50 µl) and L-Ascobric acid/VC (Sigma A0278, 110 mg/ml, 30 µl).

For HK-OP50 plates with a small amount of live bacteria, liquid cultures of bacteria were grown overnight at 37°C in LB broth. *E. coli* OP50 and *S. saprophyticus* (ATCC 15305) were diluted to the same OD$_{600}$ and 0.1 µl of the live bacteria was added onto the center of a lawn of HK-OP50 on NGM plates.

For treatment of HK-OP50 culture with enzymes, a standard overnight culture of HK-OP50 was pelleted and then re-suspended to the initial volume with S-medium. Enzymes were prepared as stock solutions in H2O and then added to the HK-OP50 in S-medium. After incubation at 37°C (210 rpm) for 90 min, about 150 µl of this enzyme-treated HK-OP50 was added to wells of a 96-well plate, where synchronized L1 larvae were seeded into each well. The source of the enzyme, stock concentration and the volume added to 1 ml of HK-OP50/S-medium are: amylase (Sigma A3403-500KU, 10–50 mg/ml, 80 µl), lipase (Sigma L1754-5G, 26 mg/ml, 80 µl) and protease (Sigma P6911-100MG, 8.2 mg/m, 80 µl).

For treatment of heat-killed OP50 with protease inhibitor, a protease inhibitor cocktail (PIC, Roche-4693124001-Complete, Mini Protease Inhibitor Cocktail Tablets) was prepared as a stock (1 tablet/ml) in H2O. 200 µl of the PIC stock solution was spread onto the top of the HK-OP50 lawn on NGM plates.

In typical experiments, eggs were added to plates immediately after the bacterial food was placed on the plate. To assay for larval growth, gonad length was measured at day 7. Three to ten replicates for each condition were performed for each experiment, and the experiments were duplicated on different days.

## Behavioral assay

For the food dwelling assay (*Kniazeva et al., 2015*), 20 µl of bacteria were spotted onto the center of 6 cm NGM plates. Synchronized L1s were seeded on the center of the lawn. After one day, animals were scored as either off the lawn or on the lawn (including those on the border). In the experiments using VB2 supplementation, 15 µl of VB2 stock solution was added on the top of HK-OP50 spot.

For the food choice assay (*Kniazeva et al., 2015*), 20 µl of bacteria were spotted onto two opposite spots on 6 cm NGM plates. The eggs were seeded on the center of plate. The number of animals at each location was scored after the time indicated in the figures. This value was divided by the total number of worms on both locations to determine the percentage in each location. In the experiments using VB2 supplementation, 20 µl of VB2 stock solution was added on the top of HK-OP50 spot.

The food-seeking assay (*Figures 3E* and *4G*) followed the protocol described in *Kniazeva et al. (2015)*. Worms were counted 3 days after placing the eggs on the origin.

Three to ten replicates for each condition were performed for each assay, and the experiments were duplicated on different days.

## Analysis of larval growth

Animals were grown on live OP50, and eggs were collected by bleaching and then washing in M9 buffer. Synchronized L1 larvae were obtained by allowing the eggs to hatch in M9 buffer for 18 hr. Eggs or synchronized L1s were seeded to plates prepared for specific assay and incubated at 20°C for 7 days. Animals were washed off the plates, mounted on agarose pads, and examined under Nomarski optics. Gonad length was measured by the ImageJ software. In some cases, larval stage was determined based on the size of the worms.

## Assay of in vivo protease activity in the worm intestinal tract

In vivo imaging of worm intestinal protease activity was performed following a published protocol (*Hama et al., 2009*) with minor modifications. Briefly, quenched BODIPY TR-X casein (EnzChek Protease Assay Kit, Invitrogen) was dissolved in 0.1 M sodium bicarbonate (pH 8.3) at a concentration of 1 mg/ml and stored at −20°C. Aliquots of EnzChek were diluted to 400 µg/ml in dH2O.

Reconstituted EnzChek was added into the tube with animals at a final concentration of 20 µg/ml and incubated for 3 hr at room temperature with shaking. Animals were then washed three times

with M9, mounted on agarose pads, and examined under a Nomarski optics. The fluorescent intensity of whole individual larvae was calculated by the Image-J software.

## qRT-PCR

For RNA isolation, *C. elegans* at the same stage (L1-L2) grown on indicated food plates were quickly collected from NGM plates and washed three times with M9 buffer, followed by the addition of 250 ul TRIzol (Invitrogen). Lysates were preserved at 80°C until RNA isolation. RNA was isolated with two chloroform extractions (50 µl each), followed by isopropanol precipitation (125 µL) of the aqueous phase, and a single wash of the resulting pellet with 70% ethanol (250 µL). RNA pellets were dried in a tissue culture hood and re-suspended in RNase-free water, and then purified of contaminating DNA by DNaseI treatment (TURBO DNA-free Kit, Invitrogen) followed by cleanup using QIAGEN RNeasy columns.

cDNA for RT-PCR experiments was synthesized using 300 ng of total RNA template, SuperScript III, and oligo(dT)12–18 primer, according to the manufacturer's protocol (Life Technologies). qRT-PCR reactions were performed in triplicate using the Applied Biosystems StepOnePlus Real-Time PCR system and Rotor-Gene SYBR Green PCR Kit (QIAGEN) and fold-change calculations were performed manually. A Student's t test was used to evaluate statistical significance. Primers can be found in *Supplementary file 1*.

## Chromatin immunoprecipitation (ChIP)

SD1963{*unc-119(ed3) III; rde-1(ne300) V;gaEx234[elt-2p::elt-2::GFP + unc-119(+)]*} worms were grown on heat-killed OP50/NGM plates and harvested. The ChIP was carried out as described in (*Mukhopadhyay et al., 2008*).

Primers can be found in *Supplementary file 1*.

## Apical protein localization in the intestine

The ERM-1::GFP (*Göbel et al., 2004*) or VHA-6::mCherry transgenic adults were bleached and eggs were collected and seeded onto indicated plates (HK-OP50, HK-OP50+VB2 or live OP50) and grown for 3 days. The apical protein localization (ERM-1::GFP or VHA-6::mCherry) was carried out according to the published method (*Zhu et al., 2015*).

## Autophagy assay

DA2123 animals carrying an integrated GFP::LGG-1 translational fusion gene (*Kang et al., 2007*), were used to quantify levels of autophagy. The DA2123 adults animals were bleached and the resulting eggs were placed on the indicated food condition (HK-OP50, HK-OP50+VB2 or live OP50). After 3 days, the animals at the equivalent stage were mounted on a 2% agar pad with a drop of M9 (plus 25 nM NaN3). Fluorescence images were captured by Nomarski microscopy. GFP-positive puncta were counted in 2–10 seam cells from approximately 50 animals in three independent experiments.

## ATP measurement

ATP level from worms was carried out as described (*Sagi and Kim, 2012*). In brief, worms grown on HK-OP50 or HK-OP50+VB2 for 3 days were collected, washed four times with M9, suspended in lysis buffer (10 mM Tris-HCl ph8.0, 25 mM NaCl, 1 mM EDTA, 1 X protease inhibitor), boiled for 20 min and quickly frozen in −80°C. All samples were processed on the same day. A Luminescent ATP Detection Assay Kit (Abcam) was used to measure ATP concentrations according to the manufacturer's instructions. ATP concentrations were normalized to absolute protein concentrations. Each assay was repeated in triplicate, and the average ATP concentration and SD were calculated.

## RNAi treatments

To analyze the effect of RNAi on growth and protease activity of worms fed HK-OP50, eggs from worms grown on OP50 were collected by bleaching, washed three times in M9 buffer, and then allowed to hatch in M9 buffer for 18 hr. The synchronized L1 worms were then transferred onto plates seeded with bacteria expressing double-stranded RNA of individual genes. After the worms grew to adults on the RNAi plate, eggs were collected again by bleaching, washed three times in M9 buffer, then seeded onto assay plates with the food to be tested. To measure larval growth,

worms grew for 7days before scoring. For the in vivo protease activity assay, worms grew for 3 days before scoring.

To screen for transcription factors that are involved in regulating protease activity in response to changes in feeding conditions by RNAi, synchronized L1 worms were transferred onto plates seeded with bacteria expressing double-stranded RNA of individual genes. The protease activity in the intestinal tract of L1-L2 worms of the next generation was measured as described above.

To assay the effect of over-expressing asp-13 and asp-14 in worms treated with elt-2(RNAi), L4 stage wild type and asp-13/asp-14 O/E worms were transferred onto plates seeded with elt-2(RNAi). Phenotypes were scored for the next generation.

For intestine-specific RNAi of asp-13 and asp-14, the strain VP303 {rde-1(ne219) V; [nhx-2p::rde-1 + rol-6(su1006)]} (Espelt et al., 2005) was used.

To assay the ATP level, synchronized L1 worms were transferred onto plates seeded with bacteria expressing double-stranded RNA of individual genes. The ATP level in L4 worms was measured as described above.

## HPLC-based analysis of flavins

Analysis of flavins from worms was carried out as described (Liuzzi et al., 2012). In brief, worms were harvested, washed in M9 buffer, suspended in lysis buffer (10 mM Tris-HCl PH8.0, 25 mM NaCl, 1 mM EDTA, 1 X protease inhibitor) and lysed by sonication. The lysate was centrifuged at 13 000 g for 1 min. Vitamin B2, FMN, and FAD content of the supernatant were analyzed by a Shimadzu Prominence modular HPLC system using a Supelcosil LC-18-THPLC column (25 cm ×4.6 mm, 5 μm particle size) (Sigma-Aldrich) following the manufacturer's instructions and detected by an inline UV/Vis detector (SPD-10A, Shimadzu). The identity of each flavin peak was determined by a retention time comparison with the relative flavin standards (Sigma-Aldrich).

## Western blot

To measure ELT-2 protein level, second generation of daf-15 RNAi-treated worms (L3 larvae arrested) or worms fed different food were analyzed by standard Western blot methods, and probed with anti-ELT-2 monoclonal antibody 455-2A4 (Wiesenfahrt et al., 2016) (Developmental Studies Hybridoma Bank, University of Iowa) and anti-Actin (Sigma-A2066) as a loading control.

## Microscopy

Analysis of fluorescent reporter expression and gonad phenotypes were performed under Nomarski optics on a Zeiss Axioplan2 microscope with a Zeiss AxioCam MRm CCD camera. Plate phenotypes were observed using a Leica MZ16F dissecting microscope with a Hamamatsu C4742-95 CCD camera.

## Statistical analysis

All statistical analyses (except quantification of the VHA-6::mCherry fluorescent signal) were performed using Student's t-test and $p < 0.05$ was considered a significant difference. Statistical analyses of the VHA-6:mCherry fluorescent signal was performed using the $\chi$ two test.

## Acknowledgements

We thank the CGC (supported by NIH P40 OD010440) for strains, Aileen Sewell for extensive discussions during the study and manuscript editing, and other lab members for advice and assistance. This work was supported by Howard Hughes Medical Institute.

## Additional information

### Funding

| Funder | Author |
|---|---|
| Howard Hughes Medical Institute | Bin Qi<br>Marina Kniazeva<br>Min Han |

The funders had no role in study design, data collection and interpretation, or the decision to submit the work for publication.

## Author contributions

BQ, Conceptualization, Data curation, Formal analysis, Validation, Investigation, Methodology, Writing—original draft; MK, Conceptualization, Data curation, Formal analysis, Investigation, Methodology; MH, Supervision, Funding acquisition, Project administration, Writing—review and editing

## Author ORCIDs

Bin Qi, http://orcid.org/0000-0003-1507-8882
Min Han, http://orcid.org/0000-0001-6845-2570

## Additional files

### Supplementary files

• Supplementary file 1. Primers used in this study.

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
