## [Decision Letter]

Thank you for submitting your article "A vitamin B2 sensing mechanism that regulates gut protease activity to impact animal's food behavior and growth" for consideration by *eLife*. Your article has been reviewed by two peer reviewers, and the evaluation has been overseen by a Reviewing Editor and Sean Morrison as the Senior Editor. The reviewers have opted to remain anonymous.

The reviewers have discussed the reviews with one another and the Reviewing Editor has drafted this decision to help you prepare a revised submission.

This is a very interesting paper that describes responses to deficiency in a critical micronutrient (Vitamin B2) in *C. elegans*. The authors show that the failure of *C. elegans* to develop on heat-killed food derives in large part from VB2 deficiency, which impairs activity of critical metabolic pathways, leading to decreased ATP production, reduced TORC1 activity, reduced expression of key intestinal digestive proteases, and increased foraging. The findings are quite novel, and will be of wide interest in the nutrition, metabolism, and TOR fields. The data are of high quality and the conclusions seem to be generally solid, but some additional points need to be addressed:

1) It is difficult to understand how worms could grow in the absence of vitamin B2, even if the asp13/14 proteases were activated by downstream intervention. Isn't VB2 required for essential biochemical reactions? Are the worms able to liberate sequestered VB2 via protease activity, or to activate a cryptic VB2 biosynthetic pathway? This seems like an issue that should be addressed (see comments 5 and 6, below). Alternatively, it's possible that the "rescued" worms only grow a little bit, but we can't tell because we are never told the size of the gonad in an adult. More on this below (specific comments #4, 5, 7).

2) How do the authors know that sensory neurons are not involved? There is precedent for sensory neuron activity affecting intestinal biology.

3) The authors say: "Further tests suggested that direct interaction with the live bacteria may be necessary for the worms to receive the benefit (Figure 1—figure supplement 1)." They show that the rescuing substance (VB2) is not volatile, but they don't show that direct contact with live bacteria is required. Conversely, it's not clear to me why, in the same section, they say that the micronutrients were provided "likely through secretion". In fact, the two sentences seem to contradict one another.

4) The authors say, in the first paragraph of the subsection “Proteases ASP-13 and ASP-14 and GATA factor ELT-2 play critical roles in VB2-promoted protease activity and food usability”, that asp-13/14 OE improved larval growth, but in Figure 3 they show only gonad length. How large is the adult gonad? How far in development do the experimental animals progress? This is a very important issue that needs to be addressed. Some clear analyses of rescue/larval staging should be performed.

5) In the last paragraph of the subsection “Proteases ASP-13 and ASP-14 and GATA factor ELT-2 play critical roles in VB2-promoted protease activity and food usability”, the effect of elt-2 OE: Again, how large is the adult gonad? How far in development do the "rescued" worms progress?

6) Is the rescue by V2B and FAD/FMN more effective than rescue by the protease and elt-2 OE (perhaps because the vitamin cofactor is provided)? Figure 5.

7) Does increasing TORC1 activity restore VB2, FAD and/or FMN levels in HK-OP50? I suspect not, but this seems like an important experiment, since growth is improved. (How can growth be improved without Vitamin B2? Same question.)

8) The results are described in terms of a pathway, but in essence the authors show that the catastrophic nutritional results of VB2 deficiency result in ATP deficiency that impairs TORC1 function and downstream events. The only compensatory feedback mechanism (which would be expected of a "sensing pathway") described is the foraging behavior. Is reduction in TORC1 activity or protease expression sufficient to lead to food-seeking behavior? The "pathway" terminology is technically correct but the simpler idea of an effect/response resulting from catastrophic nutritional failure may be more accurate.

9) Why was the ubiquitously-expressed rpl-28 promoter used for protease expression, when conclusions are made about the intestine? Is intestinal expression sufficient?

---

## [Author Response]

*[…] 1) It is difficult to understand how worms could grow in the absence of vitamin B2, even if the asp13/14 proteases were activated by downstream intervention. Isn't VB2 required for essential biochemical reactions? Are the worms able to liberate sequestered VB2 via protease activity, or to activate a cryptic VB2 biosynthetic pathway? This seems like an issue that should be addressed (see comments 5 and 6, below). Alternatively, it's possible that the "rescued" worms only grow a little bit, but we can't tell because we are never told the size of the gonad in an adult. More on this below (specific comments #4, 5, 7).*

The reviewers have raised several very interesting questions. First, VB2 level is very low but may not be entirely missing in heat-killed (HK) bacteria. To address this, we have now measured VB2 level in bacteria (in addition to the measurement in worms shown in Figure 2) and find a trace amount of VB2. These data are now added to the new Figure 2—figure supplement 2. This small amount VB2 may be enough for essential biochemical reactions to happen, and may be able to provide a small amount of ATP to support very slow and unhealthy growth under the “rescued” conditions (with over-expression of proteases or activation of TORC1). Such unwanted growth is prevented by the signaling system that senses the low level of VB2/ATP and shuts down the protease activity in wild type.

The reviewer is correct that the “rescued” growth is quite limited as the growth was very slow and the animals were very unhealthy. In fact, the “rescued” worms indicated in Figure 3 grew to late larval stages but never reach adulthood. We also measured gonad length of wild type worms at several different stages when fed normal live OP50 (new data: Figure 2—figure supplement 1) and mentioned the correspondence between the gonad length and stages in wild type. The low level of VB2 under the “rescued” condition is one reason why the “rescued” animals still grow slowly, given the broad roles of ATP in cellular processes. Moreover, as explained in the text, VB2 is only one of multiple factors missing in heat-killed bacteria, so adding back VB2 or FAD alone does not fully rescue the growth. We are developing new methods to investigate these other missing factors. However, in this study, we established the assay system to clearly detect the prominent impact of VB2.

We have modified text to clearly indicate that the “rescued” growth was slow (by days) and unhealthy. The new figures were added as mentioned above.

*2) How do the authors know that sensory neurons are not involved? There is precedent for sensory neuron activity affecting intestinal biology.*

This is a good point. We do not have any data to exclude the involvement of sensory neurons. We have modified the writing in the Discussion to point out that we do not exclude the involvement of sensory neurons in this intestinal response system.

*3) The authors say: "Further tests suggested that direct interaction with the live bacteria may be necessary for the worms to receive the benefit (Figure 1—figure supplement 1)." They show that the rescuing substance (VB2) is not volatile, but they don't show that direct contact with live bacteria is required. Conversely, it's not clear to me why, in the same section, they say that the micronutrients were provided "likely through secretion". In fact, the two sentences seem to contradict one another.*

We apologize for the poor job we did in interpreting and explaining the data regarding this issue. First, the data in Figure 1—figure supplement 1 (first 3 panels) simply suggest that the benefit was not through odorants. Second, the data from the last panel in the same figure and the data from Figure 1 suggest that worms likely obtained the benefit from a small amount of live bacteria by eating them. Finally, the data in Figure 1 may suggest that live bacteria provide some micronutrients (which led to our screening of vitamins). We now realize that our use of “physical interaction” and “secretion” were confusing and have modified the writing.

Our hypothesis was that the trace amount of live bacteria, ingested by the worms, stayed in the intestine and might continuously secrete VB2 into the gut tract, very much like microbes in mammals. Since we do not have the evidence, we will discuss this as a speculative model in this section. We have modified the writing to improve in this regard (Results part-1, second paragraph).

*4) The authors say, in the first paragraph of the subsection “Proteases ASP-13 and ASP-14 and GATA factor ELT-2 play critical roles in VB2-promoted protease activity and food usability”, that asp-13/14 OE improved larval growth, but in Figure 3 they show only gonad length. How large is the adult gonad? How far in development do the experimental animals progress? This is a very important issue that needs to be addressed. Some clear analyses of rescue/larval staging should be performed.*

*5) In the last paragraph of the subsection “Proteases ASP-13 and ASP-14 and GATA factor ELT-2 play critical roles in VB2-promoted protease activity and food usability”, the effect of elt-2 OE: Again, how large is the adult gonad? How far in development do the "rescued" worms progress?*

The reviewer has raised a very valid question. In all the rescue conditions that we described, the animals never reach adulthood. As mentioned in the text and above, HK-bacteria is missing more than just VB2.

The reason we decided to evaluate the growth by measuring gonad length was because we found that determining gonad length was a better way to determine animal growth, which is indicative of food utilization, for animals fed HK-bacteria. In other words, under poor feeding conditions, animals were very unhealthy so that stage progression may not correlate with growth well (one pathway advances while others stall). In addition, determining larval stages is less quantitative regarding measuring growth even for healthy animals. In a previous study published in *eLife* that involves staging animals (Weaver et al. 2014), one reviewer, who seemed to be an expert in the developmental timing field, specifically instructed us to measure gonad length. We believe, for our study, the key is to learn about food usage rather than defining the stage from these growth measurements.

In light of this comment, we measured gonad length of wild type worms at several different stages (new data: Figure 2—figure supplement 1) and mentioned the larval stages of VB2-fed worms based on the gonad length and vulval development in wild type. We also state clearly that animals with limited growth rescue (figure legend, growth day) by over-expressing protease or hyperactivating TORC1 never reach adulthood.

*6) Is the rescue by V2B and FAD/FMN more effective than rescue by the protease and elt-2 OE (perhaps because the vitamin cofactor is provided)? Figure 5.*

This is also a good question. The answer is yes. To compare them, we aligned all the bar graphs below (Figure 6), each with the respective control. Relative to wild type fed HK-bacteria, supplementation with VB2, FMN and FAD was more effective in rescuing growth than over expressing proteases or activating TORC1.

Author response image 1.Gonad length of worm under different conditions.**DOI:**
http://dx.doi.org/10.7554/eLife.26243.022

*7) Does increasing TORC1 activity restore VB2, FAD and/or FMN levels in HK-OP50? I suspect not, but this seems like an important experiment, since growth is improved. (How can growth be improved without Vitamin B2? Same question.)*

Good question. We have done the measurement during the review. We found that increasing TORC1 activity with the *nprl-3(ku540)* mutation does not recover the level of VB2. The new data have been added to Figure 4—figure supplement E. The reason for growth improvement is explained in our response to question #1.

*8) The results are described in terms of a pathway, but in essence the authors show that the catastrophic nutritional results of VB2 deficiency result in ATP deficiency that impairs TORC1 function and downstream events. The only compensatory feedback mechanism (which would be expected of a "sensing pathway") described is the foraging behavior. Is reduction in TORC1 activity or protease expression sufficient to lead to food-seeking behavior? The "pathway" terminology is technically correct but the simpler idea of an effect/response resulting from catastrophic nutritional failure may be more accurate.*

This is an interesting and thoughtful comment. First of all, we think we have evidence to support that the activities of the “pathway” described in this paper has a rather specific physiological role. We have added new data to show that, when live-bacteria was added to worms grown on HK-OP50 plate for 30 days, the worms recovered to adults and produced progeny, which suggest that nutrient deficiency in heat-killed food induced a protective response. This data is now presented in Figure 1—figure supplement 1. Such a role may not be consistent with the potential catastrophic nutritional failure theory.

We now also have the data to show that reducing TORC1 activity [*daf-15(RNAi*)] leads to defects in the food choice assay (new Figure: Figure 4—figure supplement 2). Perhaps more telling is that the food-seeking defect in animals fed HK-food can be significantly suppressed by activating TORC1 or specific proteases (Figure 4; Figure 3).

In addition, the data in this paper present a reasonable molecular pathway to explain the response. Previous studies have clearly established a linear relationship between VB2 and FAD, and between ATP and TORC1. We connected all of them to specific food-behavior and downstream activities. We also established the linear relationship between TORC1, *elt-2* and the expression of specific proteases. Therefore, using a pathway to describe this nutrient sensing response is still the best way. Many signaling pathways are known to have multiple inputs and outputs, as well as branch points.

Importantly, TORC1 sensing of various nutrients/chemicals including ATP has been extensively studied in mammalian cells, but few studies have connected it to animals’ response to nutrient deficiency in food or to specific physiological consequences. This study presents a good example of the usage of a common signaling pathway. Therefore, calling it a pathway has extra significance.

*9) Why was the ubiquitously-expressed rpl-28 promoter used for protease expression, when conclusions are made about the intestine? Is intestinal expression sufficient?*

This is a valid question. We use *rpl-28* is because it is not likely to be regulated by TORC1 or other known regulatory mechanisms (e.g., we learned from previous studies that it is not regulated by miRNAs). We have not expressed the genes using an intestine specific promoter since we also worry that the commonly used *ges^-1^* promoter could be regulated by TORC1 or ELT-2 which are low on heat-killed food condition. In fact, ELT-2 ChIP-seq data (Tobias et al., 2016) already shows that ELT-2 direct bind to the *ges^-1^* promoter. That why we use ubiquitously-expressed *rpl-28* promoter.

To address this issue, we performed intestinal specific RNAi by using the VP303 strain (RNAi effective only in intestine). The new data (new figure: Figure 3—figure supplement 1) shows that intestinal specific RNAi of *asp-13/14* prevents the rescue effect of VB2 supplementation on worms fed HK-food, suggesting that intestinal function of ASP-13 and ASP-14 are essential for the VB2 effect. This is consistent with the previous finding that *asp-13/14* is highly expressed in the intestine (McGhee et al., 2007) and the results from our in vitro protease/protease inhibitor assay (Figure 2).